# Implications of Inflammatory and Oxidative Stress Markers in the Attenuation of Nocturnal Blood Pressure Dipping

**DOI:** 10.3390/jcm12041643

**Published:** 2023-02-18

**Authors:** Alvaro Hermida-Ameijeiras, Nestor Vazquez-Agra, Anton Cruces-Sande, Estefania Mendez-Alvarez, Ramon Soto-Otero, Jose-Enrique Lopez-Paz, Antonio Pose-Reino, Arturo Gonzalez-Quintela

**Affiliations:** 1Department of Internal Medicine, University Hospital of Santiago de Compostela, 15706 A Coruña, Spain; 2Department of Psychiatry, Radiology, Public Health, Nursing and Medicine, University of Santiago de Compostela, 15782 A Coruña, Spain; 3Department of Biochemistry and Molecular Biology, Faculty of Medicine, University of Santiago de Compostela, 15782 A Coruña, Spain

**Keywords:** arterial hypertension, ambulatory blood pressure monitoring, dipper, oxidative stress, inflammatory markers

## Abstract

To date, no model has jointly encompassed clinical, inflammatory, and redox markers with the risk of a non-dipper blood pressure (BP) profile. We aimed to evaluate the correlation between these features and the main twenty-four-hour ambulatory blood pressure monitoring (24-h ABPM) indices, as well as to establish a multivariate model including inflammatory, redox, and clinical markers for the prediction of a non-dipper BP profile. This was an observational study that included hypertensive patients older than 18 years. We enrolled 247 hypertensive patients (56% women) with a median age of 56 years. The results showed that higher levels of fibrinogen, tissue polypeptide-specific antigen, beta-2-microglobulin, thiobarbituric acid reactive substances, and copper/zinc ratio were associated with a higher risk of a non-dipper BP profile. Nocturnal systolic BP dipping showed a negative correlation with beta-globulin, beta-2-microglobulin, and gamma-globulin levels, whereas nocturnal diastolic BP dipping was positively correlated with alpha-2-globulin levels, and negatively correlated with gamma-globulin and copper levels. We found a correlation between nocturnal pulse pressure and beta-2-microglobulin and vitamin E levels, whereas the day-to-night pulse pressure gradient was correlated with zinc levels. Twenty-four-hour ABPM indices could exhibit singular inflammatory and redox patterns with implications that are still poorly understood. Some inflammatory and redox markers could be associated with the risk of a non-dipper BP profile.

## 1. Introduction

Twenty-four-hour ambulatory blood pressure monitoring (24-h ABPM) is a better predictor of hypertension-mediated organ damage and cardiovascular disease (CVD) than isolated office measurements [1]. Blood pressure (BP) levels follow a circadian pattern characterized by a physiological decrease in nocturnal BP and individuals with an impaired circadian BP profile usually share a higher cardiovascular risk (CVR) profile [2].

Multiple cellular processes are sources of free radicals and reactive oxygen species (ROS), being superoxide anion (O2−) and hydroxyl (•OH) radicals two of the most relevant ones. Reactive oxygen species amplify oxidative processes through their reactions with divalent metals and multiple organic molecules [3].Reactive oxygen species attack polyunsaturated-fatty acids, producing intermediate lipid peroxides that propagate the oxidative process, and highly toxic end products of lipid peroxidation, such as malondialdehyde and 4-hydroxynonenal. In a pro-oxidant medium, some functional groups in the amino acids of proteins form products containing carbonyls, while reduced thiols are oxidized to disulfide bridges and other compounds disrupting the thiol–disulfide homeostasis. Fortunately, some endogenous and exogenous antioxidant molecules, such as vitamins A, E, and C, and polyphenols sustain redox balance [4,5,6].

Redox imbalance induces harmful changes in multiple organic molecules. While lipid peroxidation directly damages membrane lipids and proteins leading to fluidity and permeability impairment, protein oxidation promotes structural and functional abnormalities in some critical polypeptides and enzymes [4,7].

Reactive oxygen species, pathogen-associated molecular patterns, and pro-inflammatory cytokines, such as interleukin-6 (IL-6), interleukin-1, and tumor necrosis factor, promote some cellular pathways related to inflammation, highlighting the activation of nuclear factor kappa beta (NF-kB) protein complexes. In turn, NF-kB pathways have multiple functions related to redox balance and inflammatory responses. NF-kB activation is also responsible for the production of several inflammatory mediators, including some acute phase proteins [8].

The literature suggests that higher levels of some inflammatory markers, such as C-reactive protein (CRP), erythrocyte sedimentation rate (ESR), and fibrinogen, could represent the tip of the iceberg in the inflammatory process related to atherosclerosis [9,10].

The presence of a pro-inflammatory milieu, redox imbalance, and turbulent blood flow are factors promoting endothelial dysfunction in arterial hypertension (AHT). The oxidation of low-density lipoprotein cholesterol (LDL-c) particles in the subendothelial space transforms them into neoantigens that are recognized by some elements of the humoral immune response, whereas activation of the subendothelial macrophages is central to the inflammatory response in the vascular wall. Endothelial dysfunction and vascular inflammation are correlated with abnormalities in vascular tone in the short term and arterial stiffness in the medium–long term [11,12].

Multiple studies have evaluated the role of inflammatory and redox markers in AHT and BP circadian profile abnormalities. However, there is still no model that jointly encompasses clinical, inflammatory, and redox parameters for the prediction of a non-dipper BP profile. Moreover, little is known about the correlation between these markers and each 24-h ABPM parameter [13].

We aimed to evaluate the differences in inflammatory and redox markers between dipper and non-dipper hypertensive patients, to analyze the correlation between these markers and the main 24-h ABPM indices, and to establish a multivariate model including inflammatory, redox, and clinical markers for the prediction of a non-dipper BP profile.

## 2. Materials and Methods

### 2.1. Study Design and Participants

This was a retrospective observational study conducted at the Department of Internal Medicine of the University Hospital of Santiago de Compostela (A Coruña, Spain) between December 2021 and December 2022.

#### 2.1.1. Inclusion Criteria

We recruited patients with essential AHT older than 18 years who consented to participate. The diagnosis of AHT was established based on the results of 24-h ABPM and according to the BP thresholds and recommendations of the European Society of Hypertension (ESH) practice guidelines for office and out-of-office BP measurement [14].

#### 2.1.2. Exclusion Criteria

Clinical suspicion or diagnosis of secondary AHT, current smoking, and harmful alcohol use (considered as more than 10 g and 20 g of alcohol per day in women and men, respectively) were exclusion criteria. Diabetes Mellitus (DM) was defined as the presence of two abnormal screening test results (glycosylated hemoglobin and fasting plasma glucose) from the same sample according to the recommendations of the American Diabetes Association (ADA) guidelines [15]. Patients who met criteria for DM were excluded.

The presence of symptoms or signs of heart failure or confirmation of the diagnosis by transthoracic echocardiography performed by an expert team were exclusion criteria. Patients with high clinical suspicion of obstructive pulmonary disease or an established diagnosis after spirometric evaluation were also excluded. Renal function was assessed according to the recommendations of the chronic kidney disease (CKD) guidelines and patients with a decreased estimated glomerular filtration rate or persistent albuminuria were excluded from the study [16]. Individuals engaged in elite sports or high/very high intensity physical activity were also excluded.

We performed a detailed anamnesis and physical examination to rule out the presence of acute or chronic inflammatory processes. In cases with persistent clinical suspicion, pertinent complementary tests were performed. Patients with a diagnosis of an inflammatory process were also excluded.

### 2.2. Parameters of 24-h ABPM Collection

Patients underwent 24-hour ABPM using an oscillometric Space-Labs 90207® device (Space-Labs Inc., Redmon, WA, USA). Ambulatory BP was recorded every 20 min during the day (daytime: from 07:00 to 23:00) and every 30 min during the night (night-time: from 23:00 to 7:00). The following indices were available from the 24-h ABPM recordings: 24-h average systolic BP (24-hSBP), 24-h average diastolic BP (24-hDBP), 24-h pulse pressure (24-hPP), daytime SBP (dSBP), night-time SBP (nSBP); daytime DBP (dDBP), night-time DBP (nDBP), daytime PP (dPP), and night-time PP (nPP) [14].

nSBP decrease and nDBP dipping were calculated as the percentage (ratio between (a) daytime index minus night-time index, and (b) the daytime index) and absolute (daytime index minus night-time index) differences of the individual parameters. A non-dipper pattern was defined as a decrease in nSBP and/or nDBP of lower than 10% compared with the average daytime values. A dipper pattern was defined as a decrease in nSBP and/or nDBP equal to or higher than 10% compared with the average daytime values. The day-to-night PP gradient was calculated as the percentage (ratio between (a) daytime index minus night-time index, and (b) the daytime index) and absolute (daytime index minus night index) differences of the individual parameters. Twenty-four-hour ABPM assessment was considered reliable if more than 70% of expected measurements were valid. During the day of the test, patients filled out a form with sleeping times, drug intake, or any problems during the recording [2,14].

### 2.3. Clinical Baseline Variables

We collected information about age, sex, alcohol consumption (no versus low-risk alcohol drinking), and former tobacco use (no/yes). Body weight and height were measured in kilograms (kg) and meters (m), respectively. Body mass index (BMI) was calculated as the ratio of weight to height squared measured in kg/m^2^ [17]. Waist circumference (WC) was assessed above both iliac crests using a qualified tape and measured in centimeters (cm) [18]. We evaluated the use of antihypertensive drugs and the degree of therapeutic compliance using the Morisky–Green questionnaire [19]. The patients were asked about physical activity (moderate physical activity: Yes/No) based on the standards of the reference clinical guidelines [20].

### 2.4. Laboratory Variables. Assessment of Inflammatory Markers

Blood samples were obtained at 08:00 AM following overnight fasting of 12 h and ensuring a minimum of 12 h since the last use of antihypertensive drugs. The included variables were fasting plasma glucose (FPG), uric acid, creatinine, total proteins, total cholesterol (TC), and triglycerides (TG). All concentrations were expressed in mg/dL except for total protein levels, which were given in g/dL.

We evaluated the serum protein profile using capillary electrophoresis with a Capillarys 2 analyzer (Sebia, Lisses, France). The concentrations of alpha-, beta-, and gamma-globulins were expressed in g/dL. The alpha band includes some acute phase proteins, such as CRP. We measured wide-range CRP (wrCRP) concentrations by immunoturbidimetry in an Advia 2400 Clinical Chemistry System (Siemens, Munich, Germany) and the levels were expressed in mg/dL. The beta band includes inflammation-linked proteins such as those related to iron (Fe) homeostasis, coagulation, lipid metabolism, beta-2-microglobulin, and complement compounds. We evaluated fibrinogen levels by the Clauss method using the coagulation analyzer ACL TOP® 350 (Werfen Company, Bedford, MA, USA) and concentrations were expressed in mg/dL. Beta-2-microglobulin levels were measured by nephelometry employing a Dimension Vista® system (Siemens Healthcare Diagnostics Inc., Tarrytown, NY, USA) and concentrations were expressed in mg/dL. The gamma band mainly includes the immunoglobulins whose quantitative and qualitative evaluation allows the characterization of the immuno-inflammatory response [21,22,23,24].

Because of its relevance in inflammatory states, we included IL-6 as a pro-inflammatory cytokine. IL-6 levels were measured by a chemiluminescent immunoassay (Immulite-2000 System, Siemens Healthcare Diagnostics Inc., Tarrytown, NY, USA) and concentrations were expressed as pg/mL. Tissue polypeptide-specific (TPS) antigen levels were measured by a chemiluminescent immunoassay (Immulite-TPS; Siemens Medical Solutions Diagnostics, Gwynedd, UK) that detects cytokeratin-18 fragments, generated by cytolysis or epithelial apoptosis and concentrations were expressed in U/L. ESR is an acute phase reactant that depends mostly on the concentration of circulating acute-phase proteins, specifically fibrinogen. We measured the ESR employing an automated TEST-1 device (Alifax, Padua, Italy) that was validated using the reference Westergren method. ESR was expressed in mm/h. [25,26,27].

### 2.5. Evaluation of Redox Status: Divalent Metals, Antioxidant Vitamins, and Lipid and Protein Oxidation. Assessment of Thiobarbituric Acid Reactive Substances (TBARS) and Reduced Thiols

Serum cooper (Cu) and zinc (Zn) levels were measured using atomic absorption spectrometry with a 7700 Series ICP-MS analyzer (Agilent technologies, Inc., Santa Clara, CA, USA) and the concentrations were expressed in µg/dL. The Cu/Zn ratio was calculated as the ratio between both metals [28]. Vitamin E and vitamin A levels were measured using high-performance liquid chromatography with an Agilent Waters Acquity UPLC system (Agilent Technologies, Inc., Santa Clara, CA, USA). We adjusted vitamin A and E levels for plasma lipoprotein concentrations according to the literature. The results were expressed as µg of vitamin A and E per mg (µg/mg) of plasma lipoprotein [29].

To assess TBARS and reduced thiol levels, blood samples were collected in tubes with EDTA and centrifuged in less than 1 h after extraction at 1000 G and 4 °C for 10 min. [30]. Thiobarbituric acid forms an adduct with malondialdehyde and other lipid peroxidation products under high temperatures (90–100 °C) and acidic conditions yielding a violaceous pigment whose absorbance is directly proportional to the level of plasma lipid peroxides. We measured TBARS by spectrophotometry at 530–540 nm wavelength using an Asys UVM-340 analyzer (Biochrom®, Cambridge, UK) and following the protocol of Ohkawa, et al. The concentrations were expressed in nanomoles (nmol) per mg (nmol/mg) of plasma TC [31,32]. Ellman’s technique uses 5,5-dithio-bis-(2-nitrobenzoic acid) as a reagent to form a compound with the sulfhydryl groups of some amino acid residues in proteins, yielding a colorful pigment whose absorbance is directly proportional to the levels of reduced thiols in plasma proteins. We measured reduced thiols by spectrophotometry at 412 nm wavelength using an Asys UVM-340 analyzer (Biochrom®, Cambridge, UK). The concentrations were expressed in micromoles (µmol) per mg (µmol/mg) of plasma lipoprotein [33,34].

### 2.6. Ethics Statement

This study was conducted following the ethical principles of the Declaration of Helsinki and standards of good practice (NBP) in research of the Galician (Spain) health service (SERGAS). All patients who consented to participate provided written informed consent. The Research Ethics Committee of Santiago-Lugo approved the study protocol (code number 2021/401).

### 2.7. Statistical Analysis

Statistical analysis was developed using SPSS 22.0 statistical software (SPSS Inc., Chicago, IL, USA). We performed a descriptive analysis in which the frequencies of qualitative variables were expressed as number (*n*) and percentage (%). Quantitative variables were expressed as median and interquartile ranges (IQR).

We conducted a univariate analysis between hypertensive patients with dipper and non-dipper BP profiles. The chi-square test was used to compare categorical variables, while quantitative variables were compared using the Mann–Whitney U test. We performed a partial linear correlation analysis of some inflammatory and redox markers with the main 24-h ABPM indices ensuring the normality of the variables by a bootstrap procedure. The results were given as the Pearson correlation coefficient (r), *p*-value of the Student’s t-test for the coefficient, and standard error (SE).

We developed a binary logistic regression model for the risk of a non-dipper BP profile based on clinical, inflammatory, and redox parameters. Quantitative variables were categorized and the cutoff point was chosen based on the highest area under the receiver operating characteristic (ROC) curve (AUC) for each parameter. A *p*-value of lower than 0.1 was required for the permanence of each variable in the model. The validity and precision of the model were evaluated using the Wald test and determination coefficient (R^2^), respectively. The performance of the model was evaluated using a classification table and the results were expressed as the overall accuracy. A *p*-value of lower than 0.05 (*p* < 0.05) was considered for statistical significance.

## 3. Results

We included 247 patients (56% women) with a median age of 56 years. A total of 46 (19%) and 92 (37%) patients had low-risk alcohol consumption and were former smokers, respectively. The median values for BMI and WC were 28 kg/m^2^ and 101 cm. The most common pharmacological group was the renin–angiotensin–aldosterone system (RAAS) blockers (57%), followed by calcium channel blockers (CCBs) (42%), diuretics (28%), and B-blockers (16%).

As for the 24-h ABPM indices, SBP and DBP levels measured in mmHg are described below (expressed as median and IQR): 24-hSBP 125 (116–133), dSBP 129 (120–139), nSBP 114 (105–124), 24-hDBP 78 (71–84), dDBP 81 (74–88), and nDBP 68 (62–75). The values of laboratory variables measured in mg/dL and expressed as median and IQR were as follows: fasting plasma glucose 99 (93–109), creatinine 0.82 (0.70–0.94), uric acid 5.1 (3.9–6.3), total cholesterol 188 (166–212), and triglycerides 92 (66–131).

### 3.1. Univariate Analysis. Comparison of Baseline Variables between Dipper and Non-Dipper Hypertensive Patients

Non-dipper patients were older and had higher BMI and WC than the control group. Physical activity and therapeutic compliance were similar in both groups. However, there were differences in the frequency of diuretic use. We did not find relevant differences between the groups in the laboratory variables. All the results are shown in Table 1.

Regarding 24-h ABPM, non-dipper patients had higher levels of nDBP, nSBP, 24-hPP, and nPP than dipper individuals. Nocturnal SBP decrease and nDBP dipping also showed relevant differences between dipper and non-dipper individuals. Table 2
represents a summary of the main 24-h ABPM parameters.

### 3.2. Univariate Analysis. Comparison of Inflammatory and Redox Markers between Dipper and Non-Dipper Patients

As for the inflammatory markers, fibrinogen, wrCRP, IL-6, alpha-2-globulins, and beta-2-microglobulin levels were higher in non-dipper patients than in the control group. TPS antigen and beta-globulin levels were also higher in the non-dipper group without reaching statistical significance. We found no differences in the remaining markers. As for the redox parameters, non-dipper patients had higher levels of TBARS than dipper individuals. The Cu/Zn ratio also showed differences between the groups without reaching statistical significance. We found no differences in the remaining markers.
Figure 1 shows the differences in inflammatory and redox markers between the comparison groups.

### 3.3. Multivariate Analysis of the Risk of a Non-Dipper Pattern

We developed a binary logistic regression model based on the differences found between the comparison groups in clinical variables, inflammatory markers, and redox parameters. Age, BMI, alcohol consumption, TPS antigen, beta-2-microglobulin, TBARS, and Cu/Zn ratio showed a positive association with the risk of a non-dipper BP profile (*p* < 0.05). The results are expanded in Table 3.

After interaction analysis, we found that alcohol intake was a confounding variable for IL-6 levels, whereas BMI was also a confounding factor for wrCRP and IL-6 concentrations.

### 3.4. Linear Correlations of 24-h ABPM Indices with Inflammatory and Redox Markers

We performed a linear correlation analysis of 24-h ABPM parameters with inflammatory and redox markers adjusted for age, sex, toxic habits, WC, BMI, antihypertensive drugs, and circadian BP profile. Generally, all the BP parameters showed correlations with inflammatory and/or redox markers and such relationships were more frequent in the nocturnal BP indices. Inflammatory markers showed a higher number of correlations with 24-h ABPM indices than redox parameters. All the results are shown in Appendix A.

Alpha-1-globulins, followed by ESR, consistently showed weak positive correlations with most 24-h ABPM indices. We also found a weak positive correlation of nPP with beta-2-microglobulin (r = 0.137, *p* = 0.035, SE = 0.052) and TPS antigen (r = 0.206, *p* = 0.055, SE = 0.124). As for redox markers, vitamin E showed a positive correlation with dPP (r = 0.229, *p* < 0.001, SE = 0.059) and nPP (r = 0.178, *p* = 0.007, SE = 0.067). All the results are shown in Appendix A and Figure 2.

### 3.5. Linear Correlations of 24-h Day-to-Night BP Indices with Inflammatory and Redox Markers

We performed a linear correlation analysis of 24-h ABPM parameters with inflammatory and redox markers adjusted for age, sex, toxic habits, WC, BMI, antihypertensive drugs, and circadian BP profile. Generally, all the day-to-night indices showed some correlation with inflammatory and/or redox markers. Regarding inflammatory markers, ESR, beta-2-microglobulin, and gamma-globulins accumulated the highest number of correlations with day-to-night BP parameters. We found that nSBP dipping was negatively correlated with ESR (r = −0.120, *p* = 0.065, SE = 0.066), beta-2-microglobulin (r = −0.151, *p* = 0.007, SE = 0.068), and gamma-globulin (r = −0.391, *p*< 0.001, SE = 0.082) levels, whereas nDBP dipping was positively correlated with alpha-2-globulin (r = 0.227, *p* = 0.025, SE = 0.085) levels, and negatively correlated with gamma-globulin (r = −0.195, *p* = 0.070, SE = 0.103) and serum Cu (r = −0.129, *p* = 0.048, SE = 0.072) levels. The day-to-night pulse pressure gradient was correlated with the levels of vitamin E (r = 0.111, *p* = 0.094, SE = 0.077), serum Zn (r = 0.182, *p* = 0.005, SE = 0.054), and Cu/Zn ratio (r = −0.127, *p* = 0.052, SE = 0.063). All the results are shown in Appendix A and Figure 3.

## 4. Discussion

The results can be summarized as follows: (1) We found that higher levels of fibrinogen, TPS antigen, beta-2-microglobulin, TBARS, and Cu/Zn ratio were associated with a higher risk of a non-dipper BP profile. (2) Alpha-1 globulins and ESR showed a weak positive correlation with nocturnal systolic BP and nocturnal diastolic BP levels. (3) Nocturnal systolic BP dipping showed a negative correlation with beta-globulin, beta-2-microglobulin, and gamma-globulin levels, whereas nocturnal diastolic BP dipping was positively correlated with alpha-2-globulin levels, and negatively correlated with gamma-globulin and serum Cu levels. (4) We found a correlation between nocturnal pulse pressure and beta-2-microglobulin and vitamin E levels, whereas the day-to-night pulse pressure gradient was correlated with Zn levels.

The presence of isolated correlations between the protein electrophoretic pattern in serum and some 24-h ABPM indices could be questionable and even irrelevant. However, we systematically found such correlations for multiple parameters of BP. High levels of the alpha and gamma bands with decreased concentrations of beta-globulins, especially the beta-1 fraction, are consistent with chronic inflammatory processes. Atherosclerosis-related diseases, including AHT, are linked to chronic low-grade inflammation and might share close electrophoretic patterns [21,35].

The relationship between alpha-globulins (alpha-1 and alpha-2 fractions) and some indices of 24-h ABPM (nSBP, nDBP, and nDBP dipping) are unspecific findings and may have several readings. Chronic low-grade vascular inflammation enhances endothelial dysfunction and impairs vasomotor tone in the short-to-medium term but also increases arterial stiffness in the long term. The presence of isolated systolic AHT, lower DBP, PP widening, and excessive nDBP dipping are important features in arterial stiffness [13]. Some studies also support that low levels of some alpha-1-globulins are associated with lower BP figures and fewer cardiovascular events whereas some alpha-2-globulins have been associated with greater arterial stiffness [36,37,38].

Beta-2-microglobulin is a beta-globulin associated with high immune-cellular turnover. Its levels have also been linked to atherosclerosis load, hypertension-mediated organ damage, established kidney disease, cardiovascular events, and peripheral arterial disease, which in turn are more frequent in patients with circadian BP profile abnormalities. Although multiple studies have linked beta-2-microglobulin levels to BP figures, its possible role in abnormalities of the circadian BP profile is still poorly understood [39,40].

A pro-thrombotic state is usually an epiphenomenon more related to the presence of some CVR factors that patients with circadian BP pattern abnormalities usually share than to the BP profile itself. However, some studies along with ours have shown that, independently of some clinical variables such as age, body weight, and alcohol consumption, fibrinogen levels continue to be related to circadian BP profile abnormalities [41,42].

Persistent polyclonal gamma-globulin rise is related to a sustained immune response in situations of chronic low-grade inflammation. In the context of endothelial dysfunction, the leakage of antibodies into the subendothelial space and the presence of neoantigens, highlighting oxidized LDL-c particles, promote inflammatory responses in the vascular wall. As mentioned above, abnormalities in the vascular tone related to vascular inflammation and endothelial dysfunction might partly explain correlations between gamma-globulin levels and some BP indices [43,44].

Erythrocyte sedimentation rate showed an unspecific relationship with multiple 24-h ABPM parameters having an unfavorable effect on them. ESR represents an indirect measure of the presence of multiple cellular and humoral compounds, increasing blood viscosity. Increased blood viscosity has been associated with multiple inflammatory conditions, including atherosclerosis-related diseases such as DM or AHT. Some studies have suggested a relationship between abnormalities of the circadian BP profile and ESR, although further evidence is needed. [22,41,45].

Circulating cytokeratin-18 and its fragments are a widely endorsed marker of necrosis and apoptosis in liver diseases that is also used in the follow-up of some epithelial oncological processes. Some studies have related TPS antigen levels to the severity of inflammation and fibrosis in liver disorders such as alcoholic and non-alcoholic fatty liver diseases. Patients with abnormalities in their circadian BP profile share a higher prevalence of overweight and metabolic syndrome than the general population, which in turn are related to a higher frequency of fatty liver disease and steatohepatitis [26,46].

The risk of a non-dipper BP profile was increased in patients with higher levels of TBARS. Lipid peroxidation plays a critical role in the oxidation of LDL-c in the subendothelial space, which is critical for the development of vascular inflammation and atherosclerosis. Lipid peroxidation can also directly promote a harmful effect in endothelial cell membranes contributing to increased permeability to LDL-c particles and other molecules involved in vascular inflammation, and impairing the endothelial-dependent relaxation of the vascular smooth muscle. However, its role in the abnormalities of the circadian BP profile is not fully understood [47,48].

Some divalent metals such as Cu and Fe, by oxidation number change, catalyze favorable redox reactions leading to ROS production and amplifying oxidative stress to multiple organic molecules. However, Zn is a non-transition element with fixed oxidation number with quite opposite effects. Among its multiple functions as an antioxidant, Zn contributes to stabilizing reduced thiol groups and antagonizing transition metal-catalyzed reactions [49,50].

Pulse pressure deserves special mention due to the presence of duality in the results as both some inflammatory markers and vitamin E have been positively correlated with dPP and nPP. However, the results may not be contradictory and simply show two concomitant antagonistic phenomena since arterial stiffness is a function of inflammation-mediated vascular damage, and the presence of higher levels of vitamin E has known vasodilatory effects due in part to its high ability to inhibit lipid peroxidation [51,52].

Pulse pressure has classically been considered an independent CVR factor closely linked to isolated systolic AHT and arterial stiffness. An increasing number of studies suggest that nPP could also play an independent prognostic role in CVD. The literature supports that narrower PP levels are related to better vascular wall compliance while wider PP levels are associated with arterial stiffness. However, the role of the day-to-night PP gradient is not yet defined and it is not known whether it could have similar or different implications to nPP and dPP [51,53].

### Limitations and Strengths

This was an observational single-center study conducted in real clinical practice. The patients were white European individuals, without DM or established CVD, so the external validity of the results could be limited for other populations. Although the collection of variables was systematic, some factors involved in inflammation, redox balance, and circadian BP profile could not be considered. Clinical and laboratory variables were collected at one point and were not evaluated longitudinally. Additionally, the evaluation of variables such as physical activity or alcohol consumption with a quantitative approach may have provided more information to some results. The diary that patients filled in during the day BP was tracked did not include a detailed record of all the patients’ activities and occupations beyond aspects that were out of the routine.

The assessment of TBARS is a sensitive but less specific technique for measuring lipid peroxidation and its accuracy has also been surpassed by other methods. However, it provides a global view of lipid peroxidation and its use has been widely endorsed in CVR and oxidative stress [47,54]. Day-to-night PP gradient, nSBP dipping, and nDBP dipping are composites of other BP parameters, so some correlations with inflammatory and redox markers could be inherited from abnormalities in the previous ones. The differences in some clinical variables such as alcohol consumption, BMI and diuretic use between groups may have affected some results despite the assessment of interaction and confounding phenomena during the analysis phase.

## 5. Conclusions

Vascular inflammation and oxidative stress promote endothelial dysfunction, which is related to vascular tone impairment and abnormalities in the circadian BP profile. Multivariate models including clinical, redox, and inflammatory markers for predicting the risk of a non-dipper blood pressure profile may be of value in the evaluation of hypertensive patients. However, the influence of the inflammation-redox binomial on 24-h ABPM indices seems to be complex as BP parameters may exhibit singular inflammatory and redox patterns. This study could be the starting point for more higher quality designs in inflammation, oxidative stress, and circadian BP profile.

## Figures and Tables

**Figure 1 jcm-12-01643-f001:**
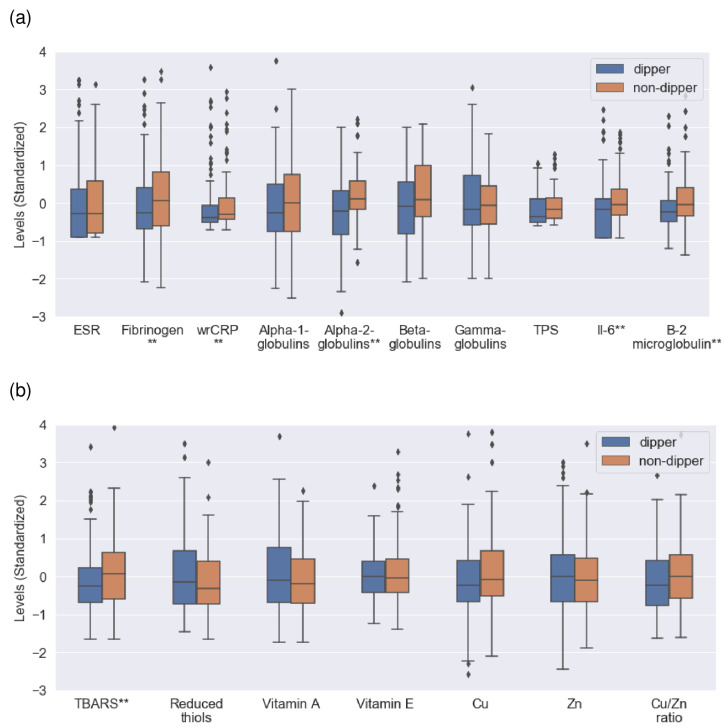
Comparison of inflammatory and redox marker levels between dipper and non-dipper patients. (**a**) Inflammatory markers (dipper and non-dipper groups, from left to right): ESR (mm/h): 8.0 (2.0–14.0), 8.0 (1.5–14.5). Fibrinogen (mg/dL): 284 (249–318), 304 (259–349). wrCRP (mg/dL): 0.12 (0.03–0.20), 0.15 (0.04–0.26). Alpha-1-globulins (g/dL): 0.29 (0.26–0.31), 0.29 (0.26–0.31). Alpha-2-globulins (g/dL): 0.73 (0.67–0.78), 0.77 (0.73–0.81). Beta-globulins (g/dL): 0.82 (0.74–0.89), 0.84 (0.76–0.91). Gamma-globulins (g/dL): 1.03 (0.87–1.18), 1.05 (0.92–1.17). TPS antigen (U/L): 33 (15–87), 55 (21–89). IL-6 (pg/mL): 2.7 (0.85–4.55), 3.2 (1.50–4.90). Beta-2 microglobulin (mg/dL): 1.74 (1.54–1.94), 1.87 (1.62–2.12). (**b**) Redox markers (dipper and non-dipper groups, from left to right): TBARS (nmol/mg TC): 2.83 (2.31–3.35), 3.25 (2.56–3.94). Reduced thiols (µmol/mg lipoprotein): 0.32 (0.20–0.44), 0.29 (0.19–0.38). Vitamin A: (mg/mg TG): 0.74 (0.40–1.07), 0.70 (0.42–0.97). Vitamin E (µg/mg TC): 16.0 (10.1–21.9), 14.6 (9.76–19.43). Cu (µg/dL): 97 (85–109), 99 (86–112). Zn (µg/dL): 98 (86–110), 96 (85–107). Cu/Zn ratio: 0.98 (0.79–1.16), 1.06 (0.88–1.24). ESR―Erythrocyte sedimentation rate. wrCRP―Wide-range C-reactive protein. TPS―Tisular polypeptide specific antigen. IL-6-―Interleukin 6. TBARS―Thiobarbituric acid reactive substances. TG―Triglycerides. TC―Total cholesterol. Cu―Copper. Zn―Zinc. All results were expressed as median and interquartile range. ** refers to *p*-value of lower than 0.05.

**Figure 2 jcm-12-01643-f002:**
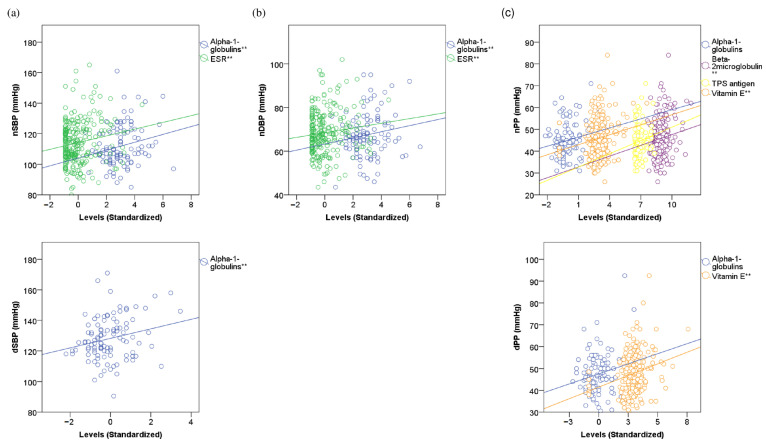
Linear correlations of 24-h ambulatory blood pressure monitoring indices with inflammatory and redox markers. The Scatter plots represent the correlations between 24-h ambulatory blood pressure monitoring indices and inflammatory and redox markers. Correlation coefficients and standard errors are shown in Appendix A. (**a**) Night-time and daytime systolic blood pressure. (**b**) Night-time diastolic blood pressure. (**c**) Night-time and daytime pulse pressure. SBP―Systolic blood pressure. DBP―Diastolic blood pressure. nSBP―Average night-time SBP. dSBP―Average daytime SBP. nDBP―Average night-time DBP. dDBP―Average daytime DBP. nPP―nocturnal pulse pressure. dPP―daytime pulse pressure. ESR―Erythrocyte sedimentation rate. TPS―Tissue polipeptide aspecific. ** refers to *p*-value of lower than 0.05.

**Figure 3 jcm-12-01643-f003:**
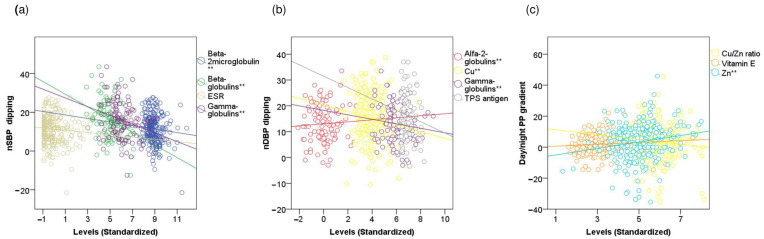
Linear correlations of 24-h day to night blood pressure indices with inflammatory and redox markers. The Scatter plots represent the correlations between 24-h day-to-night blood pressure indices and inflammatory and redox markers. Correlation coefficients and standard errors are shown in Appendix A. (**a**) Nocturnal SBP dipping. (**b**) Nocturnal DBP dipping. (**c**) Day-to-night PP gradient. SBP―Systolic blood pressure. DBP―Diastolic blood pressure. nSBP―Average night-time SBP. nDBP―Average night-time DBP. Day/night―day-to-night. PP―Pulse pressure. ESR―Erythrocyte sedimentation rate. TPS―Tissue polypeptide specific. Cu―Copper. Zn―Zinc. ** refers to *p*-value of lower than 0.05.

**Table 1 jcm-12-01643-t001:** Clinical and laboratory features. Comparisons between the groups of interest attending to the blood pressure circadian profile.

Variables	Groups ^a^		*p*-Value
	Dipper (*n* = 135)	Non-Dipper (*n* = 112)	
Age (years) †	54 (46–61)	58 (49–66)	<0.001
Sex (women) ‡	81 (60)	57 (51)	0.191
Alcohol intake ^b^‡	18 (13)	28 (25)	0.029
Former smokers ‡	54 (40)	38 (34)	0.395
Physical activity ‡	58 (43)	46 (41)	0.796
BMI (kg/m^2^) †	28 (25–31)	30 (26–34)	0.018
WC (cm) †	100 (91–108)	105 (97–112)	0.024
RAAS blockers ‡	72 (53)	68 (61)	0.300
Diuretics ‡	27 (20)	41 (37)	0.006
CCBs ‡	51(40)	49 (45)	0.515
B-blockers ‡	21 (16)	19 (17)	0.900
Compliant patients ‡	109 (81)	90 (80)	0.999
FPG (mg/dL) †	98 (89–106)	101 (92–109)	0.097
Creatinine (mg/dL) †	0.80 (0.65–0.95)	0.82 (0.72–0.92)	0.259
Uric acid (mg/dL) †	4.8 (3.6–5.9)	5.2 (4.1–6.2)	0.106
Total proteins (g/dL) †	7.2 (6.9–7.5)	7.2 (6.9–7.5)	0.294
TG (mg/dL) †	92 (58–125)	92 (60–124)	0.170
TC (mg/dL) †	188 (163–212)	186 (165–207)	0.623

^a^ Groups attending to the circadian blood pressure profile. ^b^ Low-risk alcohol consumption (defined as less than 10 g and 20 g of alcohol per day in women and men, respectively). Results expressed as † refer to median and interquartile range. Results expressed as ‡ refer to number and percentage. BMI―Body mass index. WC―Waist circumference. RAAS―Renin–angiotensin–aldosterone system. CCBs―Calcium channel blockers. FPG―Fasting plasma glucose. TG―Triglycerides. TC―Total cholesterol. Kg―Kilogram. m―Meter. cm―Centimeter. mg―Milligram. dL―Deciliter. g―Grams.

**Table 2 jcm-12-01643-t002:** Twenty-four-hour ambulatory blood pressure monitoring. Comparison of blood pressure indices between dipper and non-dipper patients.

Variables	Groups ^a^		*p*-Value
	Dipper (*n* = 135)	Non-Dipper (*n* = 112)	
24-hSBP (mmHg) †	125 (115–135)	125 (116–133)	0.978
dSBP (mmHg) †	129 (118–139)	127 (118–135)	0.171
nSBP (mmHg) †	107 (99–115)	120 (111–128)	<0.001
24-hDBP (mmHg) †	80 (73–86)	75 (68–82)	0.024
dDBP (mmHg) †	83 (76–90)	77 (70–84)	<0.001
nDBP (mmHg) †	66 (60–71)	70 (63–76)	<0.001
24-hPP (mmHg) †	45 (40–50)	48 (41–54)	0.005
dPP (mmHg) †	46 (40–52)	48 (41–54)	0.163
nPP (mmHg) †	43 (38–47)	49 (42–55)	<0.001
nSBP dipping (mmHg) †	20 (14–25)	8 (4–11)	<0.001
nSBP dipping (%) †	15 (11–18)	6 (3–9)	<0.001
nDBP dipping (mmHg) †	17 (12–21)	7 (4–10)	<0.001
nDBP dipping (%) †	20 (15–25)	9 (5–12)	<0.001

^a^ Groups attending to the circadian blood pressure profile. SBP―Systolic blood pressure. Results expressed as † refer to median and interquartile range. DBP―Diastolic blood pressure. 24-hSBP―Average 24-h SBP. 24-hDBP―Average 24-h DBP. dSBP―Average daytime SBP. nSBP―Average night-time SBP. dDBP―Average daytime DBP. nDBP―Average night-time DBP. 24-hPP―24-h Pulse pressure. dPP―daytime PP. nPP―Night-time PP. mmHg―Millimeter of mercury. %―Percentage.

**Table 3 jcm-12-01643-t003:** Binary logistic regression model based on clinical variables, inflammatory markers, and redox parameters for the presence of a non-dipper blood pressure profile.

Variables ^a,b^	B	* p * -Value	Exp(B)	95%CI	
				Inferior	Superior
Age (>63.5 years) ^c^	0.797	0.029	2.219	1.086	4.536
BMI (>33 Kg/m^2^) ^c^	1.775	<0.001	5.901	2.178	15.986
Alcohol consumption (yes)	0.937	0.022	2.553	1.146	5.688
TPS antigen (>183 U/L) ^c^	1.016	0.004	2.763	1.391	5.487
Fibrinogen (>290 mg/dL) ^c^	0.806	0.016	2.238	1.160	4.320
Beta-2-microglobulin (>1.74 mg/dL) ^c^	0.659	0.041	1.932	1.027	3.637
TBARS (>3.2 nmol/mg TC) ^c^	0.671	0.034	1.955	1.051	3.638
Cu/Zn ratio (>1.06) ^c^	1.274	0.001	3.576	1.743	7.339

^a^ A total of 247 patients entered the equation. ^b^ Only variables that reached statistical significance in the model (*p* < 0.05) are shown. ^c^ The cut-off point for the quantitative variables was chosen based on the highest area under the receiver operating characteristic (ROC) curve (AUC). The remaining variables that entered the model but did not reach statistical significance were as follows: Waist circumference, 24-h diastolic blood pressure, wide-range C-reactive protein, interleukin-6, erythrocyte sedimentation rate, and diuretics use. Model summary: *p*-value (F-test) < 0.001, −2LL = 266.303, R^2^ (Nagelkerke) = 0.346, Overall accuracy = 0.72. BMI—Body mass index. TPS—Tissue polypeptide antigen. TBARS—Thiobarbituric acid reactive substances. TC—Total cholesterol. Cu—Copper. Zn—Zinc. Kg—Kilogram. m—Meter. U—Unit. L—Liter. mg—miligrams. DL-deciliter.

## Data Availability

The data presented in this study are available on request from the corresponding author. In accordance with Article 18.4 of the Spanish Constitution and the Organic Law on Data Protection and Guarantee of Digital Rights (LOPDGDD) of 6 December 2018, the privacy and integrity of the individual will be protected at all times so anonymous data are available upon reasonable request.

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
