# Peer review of "Implications of Inflammatory and Oxidative Stress Markers in the Attenuation of Nocturnal Blood Pressure Dipping"

_jcm, 2023, doi:10.3390/jcm12041643_

Round 1
Reviewer 1 Report
Dear Sirs,
I would like to thank you very much for sending me the study entitled:
„Correlation of blood pressure circadian profile with inflammatory and oxidative stress markers in hypertensive patients. Risk assessment of a non-dipper blood pressure profile.”
I received for review an interesting, well written manuscript affecting patients with atrial hypertension. The authors analysed a total of 247 patients. The analysis was observational in nature. In addition to baseline parameters, which is the great value of this work, numerous biochemical parameters were analyzed. These parameters may, in the future, make it possible to distinguish a group of hypertensive patients with increased cardiovascular risk. The small group of patients included in the analysis may not be sufficient to draw conclusions of clinical relevance. Did the authors consider separating patient groups according to BMI (<20, 20-25, 25-30, 30-35, >35)? We know that higher body weight promotes higher activity of inflammatory markers.
Sincerely,
Reviewer 2 Report
The authors provide a novel study investigating the relationship between hypertensive patients with non-dipper pressure profiles and inflammation/oxidative stress. The authors are to be commended for their efforts in improving this field. Please see some comments below, and
Title: A bit lengthy, consider revision.
Introduction was informative and provided a level of understanding for the reader as to why and how this study can be beneficial.
Methods:
Why was physical activity, and exercise (especially prior to measurements) not accounted for? As these are major regulators of ALL markers utilized in this study. This is a weakness.
The TBARS method is one of the most volatile available (this is recognized in the limitations), and this reviewer agrees this could have been a major factor in your correlation findings.
The alcohol intake, and the difference in diuretic medications make these groups less comparable, and could be confounding the findings, could you explain this?
Please set a single level of significance for the whole of the manuscript, and do not dwell on clinical trends that a reader could misinterpret, just report the p-values and let them infer as needed.
Round 2
Reviewer 2 Report
The authors have provided an updated and improved manuscript investigating the relationship between hypertensive patients with non-dipper pressure profiles and inflammation/oxidative stress. Below are a few minor comments for the authors to address as needed.
Title: Much improved.
Line 279: Interaction-confusion? I do not know this is properly termed, perhaps leave at “interaction analysis.”
Physical activity- This is still an area of concern, it is not just physical activity habits, but what did they do the day BP was tracked? Please address this and include it in the limitation section. If this was not measured, just mention it was not.
